# Measuring the Awareness Levels of Individuals with Alcohol and Substance Use Disorders: Tertiary Prevention Standards and Development of Uskudar Result Awareness and Harm Perception Scales

**DOI:** 10.3390/brainsci13060901

**Published:** 2023-06-02

**Authors:** Nevzat Tarhan, Çiğdem Demirsoy, Aylin Tutgun-Ünal

**Affiliations:** 1Department of Psychiatry, NPISTANBUL Brain Hospital, Uskudar University, Istanbul 34662, Turkey; 2Department of Psychology Services, NPISTANBUL Brain Hospital, Uskudar University, Istanbul 34662, Turkey; cigdem.demirsoy@uskudar.edu.tr; 3Scale Development Coordinatorship, Faculty of Communication, Uskudar University, Istanbul 34662, Turkey

**Keywords:** alcohol use disorder, substance use disorder, self-awareness, rehabilitation, scale development, tertiary prevention, dangerous and harmful use, positive psychotherapy in substance use disorders

## Abstract

Rationale: Alcohol and substance use disorders are types of brain diseases that have psychological components which damage many life areas of the affected individual. Since investigating alcohol use alone is insufficient in the diagnostic evaluation process, self-awareness and the individual’s long-term psychological well-being are important in the treatment process. Primary prevention is used for preventing disease in healthy people, whereas secondary prevention is used for early diagnosis of people at risk. Tertiary prevention is important to prevent the recurrence of the disease. Since substance use disorders are a chronic problems, a new need has emerged for tertiary protection in rehabilitation standards. Methodology: In this study, we aimed to develop two scales that can provide ideas about rehabilitation standards by determining the awareness of individuals with or without alcohol and substance use disorders. By so, experts in the field can have information about the risk status of their patients in the follow-up process of rehabilitation, with the data obtained from the harm perception and result awareness dimensions in the scales. The sample consisted of 1134 participants, 41 of whom had substance use disorders. Results: Among the two scales developed in the study, the Uskudar Result Awareness Scale (USRAS) consisting of 25 items and 6 factors explained 58.4% of the total variance. The Uskudar Harm Perception Scale (USHPS), consisting of 36 items and 10 factors, explained 56.3% of the total variance. Confirmatory factor analysis of the two scales resulted in acceptable goodness-of-fit values. (*X*^2^/*df* < 3; *RMSEA* < 0.08; *NFI* > 0.90; *NNFI* > 0.95; *CFI* > 0.95; *GFI* > 0.90; *AGFI* > 0.85). Discussion: Comparisons showed that the resulting awareness of the non–SUD group was moderate (X = 3.81), whereas the SUD group had a low result awareness (X = 3.20); the effect size of the difference between the two groups was found to be high (d = 1.45; >0.8). On the other hand, the harm perception of the non–SUD group was found in the low-risk group (X = 3.78); the harm perception of the SUD group was found in the moderate-risk group (X = 3.43). According to Cohen’s d calculations, the effect size of the difference between the two groups is high (d = 1.43; >0.8). It was concluded that both of the scales are valid and safe. They can be included in the treatment process and future studies.

## 1. Introduction

Alcohol and substance use is one of the most important issues in psychiatry. Alcohol can create relief by allowing individuals to experience a certain level of pleasure. It has negative effects on the central nervous system and is the most addictive substance in the world after tobacco and caffeine [1]. Alcohol and substance use disorders are evaluated in a wide range. Investigating alcohol use alone is insufficient for diagnostic evaluation, and it is necessary to identify the problems that accompany people’s use of these substances.

In order for substance use to cause a problem for an individual and his/her environment, he/she must experience problems, uneasiness, or have impaired functionality in his/her social and/or professional life to a certain extent. Substance use is basically evaluated in three categories, including risky use, abuse, and susbtance addiction [2,3,4]. Risky use is the use of alcohol and substances that can lead to harmful consequences. This use can cause both physical and mental and social problems. Abuse is the result of not being able to assume one’s main and expected responsibilities that at work, school, or at home within a 12-month period, as defined in the fourth edition of the Diagnostic and Statistical Manual of Mental Disorders (DSM-4). However, this view changed in the current manual, which is the DSM-5. This study is in line with the DSM-5. As with epilepsy, Parkinson’s, and Alzheimer’s, substance use disorder is a brain-related disease that negatively affects a person’s functionality in daily life [5,6,7,8]. Within the framework of this study, the subject is herein discussed through the concept of substance use disorder. In this study, it we aimed to develop two scales that can provide ideas about rehabilitation standards by determining awareness in individuals with or without alcohol and substance use disorders. By so, experts in the field can have information about the risk status of their patients in the follow-up process of rehabilitation, with the data obtained from the harm perception and result awareness dimensions in the scales.

For the diagnosis of a substance use disorder, at least three of the following diagnostic criteria must be present in a way that causes clinically significant discomfort and problems for a minimum of 12 months [9]. According to DSM-5, the criteria are as follows:-Taking the substance in larger amounts or for longer than one is meant to;-Wanting to cut down or stop using the substance but not managing to;-Spending a lot of time getting, using, or recovering from the use of the substance;-Cravings and urges to use the substance;-Not managing to do what one should at work, home, or school because of substance use;-Continuing to use, even when it causes problems in relationships;-Giving up important social, occupational, or recreational activities because of substance use;-Using substances again and again, even when it puts one in danger;-Continuing to use, even when one knows that he/she has a physical or psychological problem that could have been caused or made worse by the substance;-Needing more of the substance to get the effect (tolerance);-Development of withdrawal symptoms.

It is not obligatory for a person to have tolerance and/or withdrawal symptoms to be considered to have a substance use disorder. Meeting at least three of the other diagnostic criteria is sufficient for the diagnosis. However, it is observed that there is an accompanying internal problem in most of the individuals with alcohol and substance use disorders who apply to hospitals for treatment. Many factors could accompany the disorders and, thus, they should be handled multidimensionally. In order to ensure standardization in awareness and rehabilitation studies to be carried out, model studies developed considering the biopsychosocial perspective should be updated by adding new perspectives such as positive psychology-based rehabilitation. For this purpose, current measurement tools are developed. 

### 1.1. Factors Affecting Alcohol and Substance Use Disorder

Many factors affect alcohol and substance use disorders. For example, family is considered both an etiological factor and a basic protective factor that is effectively used in the treatment against substance use disorder [1]. Accordingly, in addition to the preventive effect of positive family dynamics in alcohol and substance use disorder, familial factors such as secure attachment within the family, and close and empathic behavioral relationships are the main psychotherapeutic elements that are actively studied in the psychotherapy of individuals with substance use disorder. In order for the treatment to be completed successfully, family members and relatives who are valuable to the person should be involved in the necessary stages of the treatment. It is very important to include family members in therapy in order to prevent the emergence of a possible childhood regression in case any of the family members of the treated cases suddenly leaveing the life of the affected individual [10,11,12]. Thus, the family plays a fundamental role in strategies and techniques developed for alcohol and substance use disorders.

On the other hand, anger, profile, depression, and anxiety are associated in patients with substance use disorder [13]. As the anger level increases, it is seen that substance use has a greater effect on life, and the desire for substance use increases. For this reason, it is thought that anger is an important factor that should be evaluated in the substance use disorder process. Again, appetite, nutritional status, and quality of life are associated with individuals with alcohol and substance use disorders [14]. In a study conducted on 213 male individuals aged between 20–60 years, 167 of whom developed substance use disorder and 46 alcohol use disorder, a questionnaire was applied by face-to-face interview technique; general information, dietary habits, and 24 h food consumption status were determined. A simplified Nutritional Appetite Questionnaire (SNAQ), Dietary Diversity Score (DDS), and Short Form of Quality of Life Scale (SF-36) were administered, and anthropometric measurements were taken. As a result of the research, it was determined that 71.3% of substance users and 93.5% of alcohol users have a significant risk of weight loss in the next 6 months due to impaired appetite. In the general evaluation, it was found that individuals generally consume a single meal (57.7%), and this situation is more common in alcohol use disorder (76.1%) than in individuals using other substances (57.7%) [14].

Another aspect of the studies was to examine the sociodemographic and clinical characteristics of individuals with alcohol and substance use disorders who applied to an AMATEM (Alcohol and Substance Abuse Treatment Center) unit. In studies, it is noteworthy that the rate of seeking treatment due to alcohol and substance use in men is higher than in women; they use alcohol and substances at a level that may cause problems for their families, and there is a very high rate of job loss [15]. An in-depth interview study conducted with nine patients receiving substance use disorder treatment showed that people were exposed to labeling during the process of using substances, and faced medical, psychological, social, legal, and economic problems. The motivation to seek treatment mostly stemmed from these problems [16].

### 1.2. Rehabilitation Process in Alcohol and Substance Use Disorder

Treatment of individuals with alcohol use and non–alcoholic substance use disorders consists of medical treatment, psycho-social treatment, and social rehabilitation programs. Drug treatment is carried out in two stages: short-term and long-term. Short-term drug therapy is used to control the withdrawal symptoms that occur during the withdrawal of the addictive substance or in the physical and mental disorders that develop due to substance intoxication. Long-term drug therapy is called replacement or maintenance therapy. In this treatment, those who have developed substance use disorders such as alcohol and heroin are included in continuous and versatile treatment programs. The main purpose of social rehabilitation programs is to reintegrate individuals with substance use disorder into society and to ensure their social functionality [17,18]. Psychosocial treatments, on the other hand, are treatments in which individual therapies and group therapies are applied in which the skills are gained to prevent individuals with substance use disorder from starting to use drugs again, the behavior patterns that characterize the addiction are replaced by new behavioral patterns, and his/her relations with his/her family and environment are regulated [19]. However, more concrete and quantitative measurements can be made using standard scales consisting of factors. In this way, patients can be followed in the post-treatment period.

Substance use disorder rehabilitation has promoted treatment models that highlight chronicity rather than acuity. The biopsychosocial model is important in the treatment process of alcohol and substance use disorder [20]. 

The number of applications made due to substance use to AMATEM (Alcohol and Substance Abuse Treatment Center), which has been the institution with the highest number of applications related to alcohol and non–alcoholic psychoactive substance use in Turkey since 1983, was 78 in 1983, and this figure was 2917 in 1996. Since repeated applications are included in these numbers, when the number of drug users who applied for the first time was investigated, it was seen that it was 665 in 1993, 882 in 1994, 984 in 1995, and 955 in 1996. During the first 10 months of 2006, this number reached 1742. This situation can be considered as an indicator of the increasing problem of substance use [21].

On the other hand, it has been reported that 50–60% or more of patients with alcohol and substance use disorders started drinking again within a few months after detoxification due to negative emotional states, interpersonal conflicts, and social pressure [22,23,24,25,26,27,28,29]. Thus, while determining awareness and rehabilitation standards, it is important to structure these standards to protect the person as much as possible from relapse. At this stage, it is useful to include the perspective of positive psychology in the rehabilitation process. Positive psychology distinguishes itself from known psychology teachings. The purpose of positive psychology is not to replace conventional psychology, but to become an important complement to it. Positive psychotherapy, on the other hand, takes its source from positive psychology [30].

It is known that more than half of physical diseases are caused by living without nutrition, not exercising, and not paying attention to hygiene, that is, they are related to the consequences of lifestyle. It would not be wrong to say that more than half of mental illnesses are caused by not being able to live right and wrong life philosophy, that is, mental well-being is directly proportional to positive life philosophy. At this point, positive psychology suggests using core values and developing some skills. The science of positive psychology tries to understand and intervene in order to increase the life satisfaction and happiness of both healthy individuals and the clinical population [31,32,33,34,35]. “Positive Activity Interventions” (PAI) encourage positive emotions, positive thoughts, and/or positive behaviors rather than directly aiming to correct negative or pathological feelings, thoughts, and behaviors. It is also defined as a relatively short, self-administered, and non–stigmatizing activity or exercise. 

Results of a pilot study of eight sessions of positive intervention with a group of adolescents with substance use problems in the UK showed that groups at greater risk were more likely to benefit from positive interventions. The results of this study provided support for investigating the value of such interventions among people being treated for substance use disorders. A more detailed description of positive psychology’s contributions sheds light on its potential for substance use disorder research in general and rehabilitation in particular. The conceptual map drawn by the founding positive psychologists is in three areas: positive emotion, personality strength, and positive knowledge. These three domains are associated with three “types of happiness”. “Pleasant life” is filled with positive emotion, “good life” represents the strength of character required for full commitment, and “meaningful life” represents positive institutions that allow individuals to develop and contribute. Thus, positive psychology-based scales can be developed to include these three areas in positive psychology. The scales developed in this study are also based on this perspective.

### 1.3. Self-Awareness in Substance Use

Self-awareness of the individual is an important factor affecting the treatment process. Mindfulness is a mind and body practice that involves focusing attention on momentary experiences and observing inner experiences [36,37]. Mindfulness is defined as accepting and evaluating the positive and negative situations as they are and being aware of them [38]. In the literature, it is known that mindfulness can create a positive mood, decrease depressive relapses, increase empathy, decrease substance use, increase motivation, increase students’ academic success, decrease stress levels, increase awareness of emotions, and decrease anxiety levels [39]. The existence of three pillars of substance use disorder can be questioned as support for awareness studies. The first pillar is matter itself. The second is the substance use disorder subculture the person is in. The third pillar is the personality structure of the individual. Here, subculture is the culture that develops without breaking its connection with the dominant culture, but by separating at various important points (different purposes, expectations, behavior, attitude, actions, clothing styles, language, values, and lifestyle) [40]. When dealing with substance use disorder, it is necessary to evaluate these three dimensions together and to plan together in treatment.

When the literature is examined, scales for substance use disorder awareness and studies using scales can be found [41,42,43]. One of these scales is The Substance Addiction Awareness Scale, developed by Özay Köse and Gül in 2018 to measure substance use disorder awareness. It was developed by applying to 230 secondary school students. It consists of four factors (Support and legal regulations, Symptoms and effects of substance use, Personal attitudes and opinions, and Factors causing addiction) and 27 items. The explained variance rate of the scale was 4.9%, and the internal consistency coefficient was found to be 0.88 [42]. It is important to know the risk factors in the substance use disorder treatment process. It is important for the person to be self-aware, that is, to know themselves [44]. For example, mindful awareness is used in the treatment of substance use disorder in body-focused therapy. The treatment results of 61 women who received this body-focused therapy for eight sessions per week showed moderate to large effects [45]. It is also stated that mindfulness and regulating emotions are predictive of substance use disorder and other non–substance-related behavioral addictions [46]. Studies conducted with students in which awareness and level of knowledge are determined [47] are important as well, as it is essential that current perspectives on mindfulness be included in the measurement tools used. It is stated that there is a need for awareness scales that can be applied especially to individuals with substance use disorder [48]. In this study, we aimed to develop two scales that can provide ideas about rehabilitation standards by determining awareness in individuals with or without alcohol and substance use disorders.

## 2. Materials and Methods

### 2.1. Ethical Approval

This study received ethical approval from the Uskudar University Non–Interventional Research Ethics Committee, report number 61351342/December 2022-47 (28 December 2022). This study was performed according to the principles set out by the Declaration of Helsinki for the use of humans in experimental research.

### 2.2. Participants

In the study, the sample consisted of 1134 people over the age of 18, including the non–SUD group (n = 1091) and the SUD group (n = 43). The validity and reliability studies of the Uskudar Result Awareness Scale (USRAS) and the Uskudar Harm Perception Scale (USHPS) were conducted on a sample consisting of two groups. Subsequently, psychometric properties of the scales were determined and group comparisons were performed.

Considering the genders of the participants in the study, there were 80.8% women (n = 882) and 17.7% men (n = 193) participants in the non–SUD group; and 14% women (n = 6) and 86% men (n = 37) in the SUD group. The SUD group consisted of individuals with alcohol and substance use disorders who were hospitalized in the clinic with a mean age of 29.5 years. On the other hand, the non–SUD group consisted of volunteers over the age of 18, whose mean age was 34.6, and who were reached via the Internet. In general, the ages of the participants in the study group ranged from 18 to 67.

### 2.3. Data Collection Tools

The questionnaire which included the two scales (USRAS and USHPS) and a demographic information form were used as data collection tools in the research.

#### 2.3.1. Demographic Information Form

In the demographic information form, the participants were asked questions such as gender, age, educational status, marital status, having a child, and daily use of social media, as well as whether they used harmful substances and the frequency of sleep disorders.

#### 2.3.2. Uskudar Result Awareness Scale (USRAS) and Uskudar Harm Perception Scale (USHPS)

For the validity and reliability studies of Uskudar Result Awareness Scale (USRAS) and Uskudar Harm Perception Scale (USHPS) developed in this study, content validity, factor analysis, construct validity, discriminant validity, and internal consistency reliability were performed. Firstly, the researchers searched the literature for both scales and created an item pool by considering the principles of positive psychology in the study. Six experts were consulted for their opinions. With the expert evaluation inventory, questions in each candidate scale were evaluated as “It is appropriate for the item to remain on the scale”, “The item may remain on the scale, but unnecessary”, and “It is not appropriate for the item to remain on the scale”. A pool of experts was formed with two academicians from each of the fields of psychology, psychiatry, and communication in order to include interdisciplinary views. Inventories were sent to the experts via e-mail. Subsequently, the compatibility ratios of the items were calculated with the help of the formula proposed by Miles and Huberman [49].

Compliance rates were calculated for each item using the ratings in the inventory. Accordingly, care was taken to ensure that the relevant item was scored between 0 and 1 and not below 0.80. In addition, each item was revised and arranged in terms of spelling and grammar, taking into account the section in which the experts expressed their opinions. Thus, the 35-item Uskudar Result Awareness Scale (USRAS) and the 44-item Uskudar Harm Perception Scale (USHPS) candidate scale forms were prepared in a five-point Likert-type scale (from Strongly Disagree to Totally Agree), and the data collection phase was started for factor analysis.

Explanatory Factor Analysis is one of the statistical calculation techniques performed in accordance with a large number of variables and is frequently applied within the scope of the construct validity of scale development. Before performing EFA, it is necessary to test whether the data set is suitable for factor analysis [50]. For this, the Bartlett Test of Sphericity and Kaiser–Meyer–Olkin (KMO) test were applied to the obtained data. A KMO value of 0.90 and above is “excellent”, between 0.80 and 0.89 is “very good”, between 0.70 and 0.79 is “good”, between 0.60 and 0.69 is “moderate”, between 0.50 and 0.59 is “weak”, and below 0.59 is considered “unacceptable”. Furthermore, the Bartlett Sphericity value is expected to be significant [51,52].

When the construct validity phase of the scales proceeds, the number of factors can be determined with the EFA. For this, the Eigenvalues are used. According to the Eigenvalues, factors with this value equal to or greater than 1 are considered significant, and when it is less than 1, they are not taken into account [53]. Again, the explained variance rate revealed by factor analysis varying between 40% and 60% is ideal in social sciences. On the other hand, in the criterion validity studies of each scale, correlation values are examined in terms of the relationships of the scales with each other. It is stated that when interpreting the correlation values, a correlation value between 0.30 and 0.70 is “medium”; values above 0.70 indicate a “high” relationship and values below 0.30 indicate a “weak” relationship [51].

Confirmatory Factor Analysis is carried out to determine whether the parameters revealed by factor analysis confirm the scale structure and the goodness-of-fit values are calculated with the structural equation modeling. Goodness-of-fit values are expected to be within acceptable ranges in the literature. During the reliability studies, internal consistency analysis for items is performed according to the item variances of the scales, and Cronbach’s Alpha coefficients are calculated.

### 2.4. Criteria for Inclusion/Exclusion

While creating two groups in the study, patients who were diagnosed with alcohol use disorder and substance use disorder (American Psychiatric Association DSM-5 diagnostic criteria manual) and who received inpatient treatment in the clinic were included in the SUD group in the study. The diagnostic criteria for the non–SUD group were ignored. Persons under 18 years of age were not included in the study for the groups.

### 2.5. Procedures

Pilot Application: The comprehensibility of the questions was tested by applying the online questionnaire, which was prepared as a data collection tool in the research, to 15 people for trial purposes. After the questionnaire was administered to the undiagnosed participants, a trial application was made to two patients who were treated under the supervision of a psychologist in consultation with a psychiatrist in the AMATEM unit. It was determined that no problem was encountered during the pilot application, and then the field application was started.

Application of Scales: The online questionnaire including the Demographic Information Form, USRAS, and USHPS scales was applied to both groups on a voluntary basis for a month on 1–30 January 2023, after the approval of the Ethics Committee dated 30 December 2022. For the participants in the non–SUD group, the questionnaire was shared on digital platforms, via e-mail and SMS. For the SUD group, data were collected by applying the online survey on a PC under the supervision of a psychologist on a voluntary basis.

### 2.6. Data Processing and Statistical Analysis

Exploratory factor analysis (EFA) was applied in the USRAS and USHPS construct validity studies. The correlation coefficient of the Pearson product of moments was calculated for the relationship between the dimensions of the scales and the sum of the scales. Pearson Correlation Coefficient test was applied in criterion validity studies. The internal consistency reliability coefficient of the scales was determined by the Cronbach Alpha value. In the confirmatory factor analysis study, goodness-of-fit values (*X*^2^/*df*, *RMSEA*, *NFI*, *NNFI*, *CFI*, *GFI*, *AGFI*) were evaluated in the presence of a structural equation model on a 400-person data set, which consists of groups classified as non–SUD and SUD. While comparing the non–SUD group and SUD group, the normal distribution was checked in the data set. In the case of normal distribution, the differentiation of independent variables according to the dependent variable was examined by parametric tests (independent group *t*-test and one-way analysis of variance). SPSS 26.0 and AMOS statistical program were used for all validity and reliability analyses and comparison tests.

## 3. Results

### 3.1. The Validity and Reliability Studies of the Uskudar Result Awareness Scale (USRAS) 

At the beginning of the validity and reliability studies, whether the data were suitable for factor analysis was examined with the Kaiser–Meyer–Olkin (KMO) sampling coefficient and the Bartlett Test of Sphericity. Accordingly, the KMO coefficient value was found to be 0.85. The result of the Bartlett Test of Sphericity was found to be significant (X^2^ = 9271.572; df: 300; *p* = 0.000). Thus, the results showed that the data were suitable for factor analysis (52). Exploratory factor analysis (EFA) was performed with 35-item candidate scale data created after the expert opinion phase. During EFA, Eigenvalues greater than 1 formed a factor and a six-factor structure emerged for USRAS [53].

As seen in Table 1, the Eigenvalues of the factors varied between 5.38 and 1.14. The explained variance rate in the total scale was 58.49%. After determining the number of factors, item factor loads were examined, and when the factor load of each item was 0.50, a suitable structure with 6 factors and 25 items emerged. Thus, 10 items (7, 9, 10, 13, 14, 17, 20, 21, 22, 32) in the scale were eliminated from the scale due to their presence in more than one factor or low factor load. The factor loading values of the items are given in Table 2.

As seen in Table 2, the factor and item distributions in the scale were determined, and the items were renumbered. Subsequently, each of the factors to which the new numbered items belong was given a name. Accordingly, Factor 1 (Items 1–7) is “*Medium and Long-Term Plan*”; Factor 2 (Items 8–11) is “*Action*”; Factor 3 (Items 12–15) is “*Decision*”; Factor 4 (Items 16–18) is “*Short-Term Plan*”; Factor 5 (Items 19–22) is “*Emotion/Intention*”; and Factor 6 (Items 23–25) was named as “*Philosophy of Life*”. Item-total correlations were found to be within the acceptable range for each item and correlated with the scale (r > 0.30). Within the scope of reliability studies, Cronbach Alpha values, which are the internal consistency coefficients of the factors and the scale total, were calculated. The Cronbach Alpha values of the factors were found to be between 0.67 and 0.82, and the total scale was found to be 0.72.

After the USRAS factor structure was formed, a scale model was drawn using the AMOS program with the data set consisting of 400 people, and confirmatory factor analysis was applied. Whether the model was validated or not was tested with goodness-of-fit values (Figure 1). When the model was run by creating covariance among the factors in the model, the goodness-of-fit values were found to be in the acceptable range and the model was validated (*X*^2^/*sd* = 2.47 < 3; *RMSEA* = 0.06 < 0.08; *NFI =* 0.92 > 0.90; *NNFI* = 0.96 > 0.95; *CFI* = 0.97 > 0.95; *GFI* = 0.91 > 0.90; *AGFI* = 0.87 > 0.85).

### 3.2. Intergroup Awareness Scores of Uskudar Result Awareness Scale (USRAS)

As a result of the validity and reliability studies, the average scores of the participants from the scale were calculated by adding each item and dividing it by 25. The mean scores of the non–SUD and SUD groups in the study were also obtained using the Effect Size (d) calculation developed by Cohen (1988) [54]. The effect size of the groups was revealed and is shown in Table 3.

The Non–SUD and SUD groups were taken as references while performing the statistical analysis. In the first stage, the scale scores of the total of the non–SUD group and the total of the SUD group were compared with the independent group t-test, and the result was found to be significant (t = 9.17; *p* < 0.001). It was revealed that the SUD group’s result awareness was lower than the non–SUD group (X = 3.20). The effect size of the difference between the SUD group and the non–SUD group was evaluated with Cohen’s d effect size calculation, taking into account the scale scores. Accordingly, the non–SUD group was found to be in a higher impact area in terms of result awareness compared with the SUD group (d = 10.45; >0.8). 

When evaluating the effect size of Cohen, it is stated that if the d value is less than 0.2, the effect size can be defined as “weak”; if it is 0.5, it can be defined as “medium”, and if it is greater than 0.8, it can be defined as “strong” [54]. In addition, when the difference between the two groups according to gender was examined, the number of men and women was found to be suitable for the non–SUD group statistically. In the independent group *t*-test calculations, there was no difference according to gender (t = 1.93; *p* > 0.005). In the SUD group, no comparison was made since the number of women was less than 30 (Figure 2).

The difference between the groups and the effect size was analyzed in the comparison of the scale scores of the non–SUD and SUD groups for harmful substance use (Table 4). 

First, the non–SUD group was compared with a one-way ANOVA analysis according to the three categories within itself, and the difference between the groups was found to be significant (*p* < 0.001). When the difference between the groups was examined by LSD analysis, the resulting awareness of the group that replied “never use” was found to be the highest compared with the other groups. The result awareness score of the group with “tobacco, alcohol and substance use” was found to be the lowest compared to the other two groups (X = 3.47). The difference in the three categories within the non–SUD group was also examined by Cohen’s d effect size calculation. The group that never used harmful substances was found to be in a higher effect area compared with the group that “uses tobacco, alcohol and substance” (d = 0.82; >0.8). The group that never used was slightly above the weak effect area and at the border of the medium effect area compared with the “only tobacco use” group (d = 0.24; >0.2 < 0.5). The “only tobacco use” group included in the non–SUD group was found to be in the medium effect area compared with the “tobacco, alcohol and substance use” group (d = 0.59; >0.5). Lastly, the “tobacco, alcohol and substance use” group in the non–SUD group and the SUD group (tobacco, alcohol and substance use) were compared with the independent group *t*-test, and the difference was found at the border (*p* = 0.05). According to the effect size calculation, the affected area of the group using more harmful substances in the non–SUD group was found to be moderate compared to the SUD group using the same type of harmful substances” (d = 0.57; >0.5). The scale score of the SUD group was found to be the lowest in all categories in the non–SUD group (X = 3.20). In other words, the result awareness level was observed to be the lowest.

### 3.3. The Validity and Reliability Studies of the Uskudar Harm Perception Scale (USHPS) 

At the beginning of the validity and reliability studies, whether the data were suitable for factor analysis was examined with the Kaiser–Meyer–Olkin (KMO) sampling coefficient and the Bartlett Test of Sphericity. Accordingly, the KMO coefficient value was found to be 0.85. The result of the Bartlett Test of Sphericity was found to be significant (X^2^ = 10,198.805; df: 630; *p* = 0.000). Thus, the results showed that the data were suitable for factor analysis. After receiving expert opinions on the scale, exploratory factor analysis (EFA) was applied to the data collected with the 44-item candidate scale. During EFA, values with Eigenvalue greater than 1 formed a factor and a 10-factor structure emerged for USHPS [53].

As seen in Table 5, the Eigenvalues of the factors varied between 6.37 and 1.02. The explained variance rate in the total scale was 56.36%. After determining the number of factors, item factor loads were examined, and when the factor load of each item was 0.50, a suitable structure with 10 factors and 36 items emerged. Thus, 8 items (3, 13, 19, 21, 22, 33, 35, 38) in the scale were eliminated from the scale due to their presence in more than one factor or low factor load. The factor loading values of the items are given in Table 6.

As seen in Table 6, the factor and item distributions in the scale were determined, and the items were renumbered. Subsequently, each of the factors to which the new numbered items belong was given a name. Accordingly, Factor 1 (Items 1–6) is “*Objectivity and Long-Term Perceptual Blindness*”; Factor 2 (Items 7–11) is “*Stress Relief*”; Factor 3 (Items 12–156) is “*Impulsivity and Subjective Reality Blindness*”; Factor 4 (Items 17–19) is “*Curiosity*”; Factor 5 (Items 20–22) is “*Harm Avoidance*”; Factor 6 (Items 23–24) is “*Sensation-seeking*”; Factor 7 (Items 25–27) is “*Perception of Narcissism*”; Factor 8 (Items 28–31) is “*Perception of Hedonism*”; Factor 9 (Items 32–33) is “*Control*”; and Factor 10 (Items 34–36) was named as “*Intention*”. Item-total correlations for USHPS were within the acceptable range for each item and correlated with the scale (r > 0.30). Within the scope of reliability studies, Cronbach Alpha values, which are the internal consistency coefficients of the factors and the scale total, were calculated. The Cronbach Alpha values of the factors were found to be between 0.68 and 0.86, and the total scale was found to be 0.83.

After the USHPS factor structure was formed, a scale model was drawn using the AMOS program with the data set consisting of 400 people, and confirmatory factor analysis was applied. Whether the model was validated or not was tested with goodness-of-fit values (Figure 3). When the model was run by creating covariance among the factors in the model, the goodness-of-fit values were found to be in the acceptable range and the model was validated *(X*^2^*/df* = 2.01 < 3; *RMSEA* = 0.05 < 0.08; *NFI* = 0.93 > 0.90; *NNFI* = 0.97 > 0.95; *CFI* = 0.96 > 0.95; *GFI* = 0.92 > 0.90; *AGFI* = 0.86 > 0.85). 

### 3.4. Intergroup Awareness Scores of Uskudar Harm Perception Scale (USHPS) 

As a result of the validity and reliability studies, the average scores of the participants from the scale were calculated by adding each item and dividing it by 36. By using the Effect Size (d) calculation developed by Cohen (1988) [54], the effect size of the groups was revealed and is shown in Table 7.

The Non-SUD and SUD groups were taken as references while performing the statistical analysis. In the first stage, the scale scores of the total of the non–SUD group and the total of the SUD group were compared with the independent group *t*-test, and the result was found to be significant (t = 7.81; *p* < 0.001). It was revealed that the SUD group’s harm perception was lower than the non–SUD group (X = 3.43). The effect size of the difference between the SUD group and the non–SUD group was evaluated with Cohen’s d effect size calculation, taking into account the scale scores. Accordingly, the non–SUD group was found to be in a higher impact area in terms of harm perception compared with the SUD group (d = 1.43; >0.8). In addition, when the difference between the two groups according to gender was examined, the number of men and women was found to be suitable for the non–SUD group statistically. In the independent group *t*-test calculations, there was no difference according to gender (t = 1.55; *p* > 0.005). In the SUD group, no comparison was made since the number of women was less than 30 (Figure 4).

The difference between the groups and the effect size was analyzed in the comparison of the USHPS scale scores of the Non–SUD and SUD groups for harmful substance use (Table 8). 

For USHPS score comparisons, three categories within the non–SUD group were compared with one-way ANOVA analysis, and the difference between groups was found to be significant (*p* < 0.001). When the difference between the groups was examined by LSD analysis, the harm perception of the group that replied “never use” was found to be the highest compared with the other groups. The harm perception score of the group with “tobacco, alcohol and substance use” was found to be the lowest compared with the other two groups (X = 3.51). The difference in the three categories within the non–SUD group was also examined by Cohen’s d effect size calculation. The group that never used harmful substances was found to be in a higher effect area compared with the group that “uses tobacco, alcohol and substance” (d = 1.02; >0.8). The group that never used was slightly above the weak effect area and at the border of the medium effect area compared with the “only tobacco use” group (d = 0.28; >0.2 < 0.5). The “only tobacco use” group included in the non–SUD group was found to be in the medium effect area compared with the “tobacco, alcohol and substance use” group (d = 0.69; >0.5). Lastly, the “tobacco, alcohol and substance use” group in the non–SUD group and the SUD group (tobacco, alcohol and substance use) were compared with the independent group *t*-test, and no difference was found (*p* > 0.05). According to the effect size calculation, the affected area of the group using more harmful substances in the non–SUD group was slightly above the low effect compared with the SUD group using the same type of harmful substances (d = 0.24; >0.2 < 0.5). This shows that there is not much difference in the harm perception between the non–SUD and SUD groups, where the use of harmful substances is most intense.

### 3.5. Criterion Validity of Scales

Since the USRAS and USHPS scales were thought to be related, the Pearson Correlation (r) correlation coefficient of these two scales was calculated for criterion validity. As expected, a relationship was found between the USRAS scale and the USHPS scale. As in Table 9, it is seen that this relationship is of medium strength and this relationship is significant (r = 0.54; *p* < 0.001). 

## 4. Discussions

The prevalence of alcohol and substance use disorder in Turkey as well as all over the world necessitates the review of current rehabilitation studies on the subject and the discovery of new standards. Psychometric scales constitute a part of these processes based on concrete data. With these data, experts in the field can draw conclusions about people’s self-awareness levels. Rehabilitation is definitely the ultimate goal of treatment; however, it is not the treatment itself. Relapse is related to addiction, whereas awareness of harm is somewhat irrelevant. People with alcohol and substance use disorders can be aware of a substance’s dysfunctional use, although they can be unaware of its nature, mechanism, and core issues (chronicity, automatic relapsing, and autonomy with respect to environment). Improving awareness is a way to prevent relapse, as long as this relies on ongoing treatment.

In this study, two psychometric scales named “Uskudar Result Awareness Scale” (USRAS) and “Uskudar Harm Perception Scale” (USHPS) were developed using a data set of 1134 people, consisting of individuals with or without substance use disorder aged 18 and over. In addition, for both scales, a model was created from factor structures using the AMOS program in the data set of 400 people (non–SUD group + SUD group), and acceptable goodness-of-fit values were obtained by confirming the model with confirmatory factor analysis. Accordingly, USRAS, consisting of 25 items and 6 factors, explained 58.49% of the total variance. The internal consistency reliability Cronbach Alpha value was found to be 0.72 on the total scale. The USHPS scale consisted of 36 items and 10 factors and explained 56.36% of the total variance. The internal consistency reliability Cronbach Alpha value was found to be 0.83 on the total scale.

In the first measurement performed in the study, the USRAS scale score of the sample of 1134 people was found to be 3.81 points for the non–SUD group and 3.20 points for the SUD group. Evaluation score ranges of the scale were calculated using the equal spacing technique, taking into account at least 1 and a maximum of 5 for each item (paying attention to the items to be reverse coded). According to this, between 25 and 58 points is evaluated as “Low Result Awareness”; between 59 and 89 points is evaluated as “Medium Result Awareness”; and between 90 and 125 points is evaluated as “High Result Awareness”. As a result of multiplying the scores obtained by dividing 25 by 25 in the study, the resulting awareness of the non–SUD group was moderate (X = 95.25). It was found that the SUD group had a low result awareness (X = 80.00). According to Cohen’s d calculations, the effect size of the difference between the two groups was found to be high (d = 1.45; >0.8). Since the SUD group consisted mostly of men and the total number was not high (n = 43) in the study, the difference between men and women could not be analyzed statistically. However, there was no difference between men and women in the non–SUD group.

In USHPS measurements, the score of the non–SUD group was found to be 3.78, and the score of the SUD group was found to be 3.43. Evaluation score ranges of the scale were calculated using the equal spacing technique, taking into account at least 1 and a maximum of 5 for each item (paying attention to the items to be reverse coded). According to this, between 36 and 83 points is evaluated as “Low harm perception/very high risk”; between 84 and 132 points is evaluated as “Moderate harm perception/moderate risk”; and between 133 and 180 points is evaluated as “High harm perception/low risk”. As a result of multiplying the scores obtained by dividing 36 by 36 in the study, the harm perception of the non–SUD group was found as high, and in the low-risk group (X = 136.08); the harm perception of the SUD group was found in the moderate risk group (X = 123.48). According to Cohen’s d calculations, the effect size of the difference between the two groups was found to be high (d = 1.43; >0.8).

Finally, according to the results obtained for the non–SUD group in the triple category (1: Never use, 2: Only tobacco use, 3: Tobacco, alcohol and substance use), the resulting awareness and harm perception decreased as the use of harmful substances increased. The group that did not use any of them had the highest awareness of consequences and perception of harm, followed by the group that only used tobacco. The group that used tobacco, alcohol, and other substances had the lowest levels of result awareness and harm perception. The SUD group comprised of the group that currently uses tobacco, alcohol, and substances, and is in any case at a low level in terms of result awareness and harm perception compared with the non–SUD group. Validity and reliability studies and fit index values of the two scales developed in the study were found as acceptable. The strengths of the study are that the scales were developed in a study group that included non–SUD and SUD groups and that their usability was tested by giving functional results in the first comparisons. On the other hand, there is a need to expand the study by ensuring the proportionality of the number of men and women among SUD groups and with larger samples where the number of individuals with alcohol and substance use disorder is also higher. The potential of testing the effectiveness of the scales by including the scales is important. For follow-up of the treatment process and to prevent relapse, creating self-awareness should not be ignored.

In summary, this study proved that these scales are capable of measuring and evaluating whether people “know the natural consequences of their actions or whether they realize that they are living harmful and dangerous” with their sub-dimensions, by comparison with healthy control in this study. However, the main goal of this study was to develop scales for measuring awareness. The scale development is emphasized more in this study, rather than discussing the research findings with the current literature. With the scales applied, different risk groups emerged as low, medium, and high. When the treatment of the addiction is finished, measuring awareness levels can give the experts in the field an idea that is based on data. Patients can be followed in a standardized manner by means of the factors in the scales. Treatment programs can be created for different risk groups. This is a suggestion of the research and can be carried out in future studies.

## Figures and Tables

**Figure 1 brainsci-13-00901-f001:**
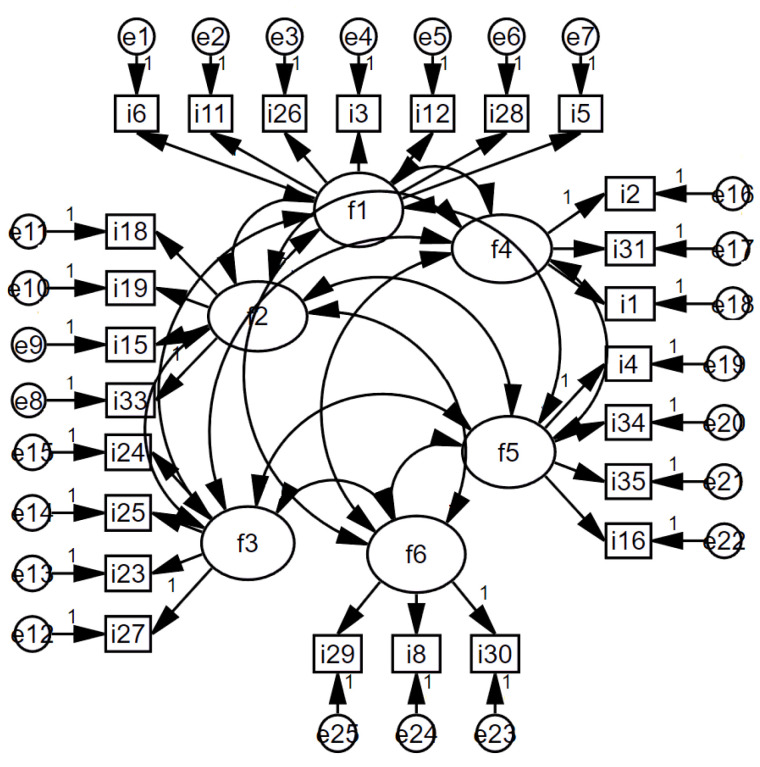
USRAS Scale Standardized Model.

**Figure 2 brainsci-13-00901-f002:**
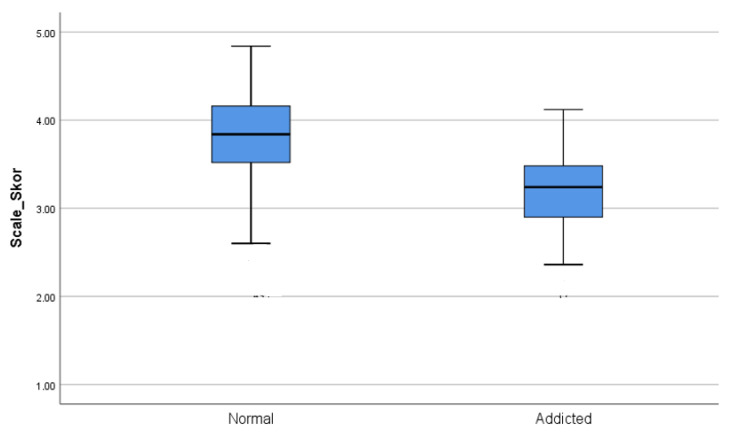
USRAS Scale Scores of the groups (the cut-off value was accepted as 2.5).

**Figure 3 brainsci-13-00901-f003:**
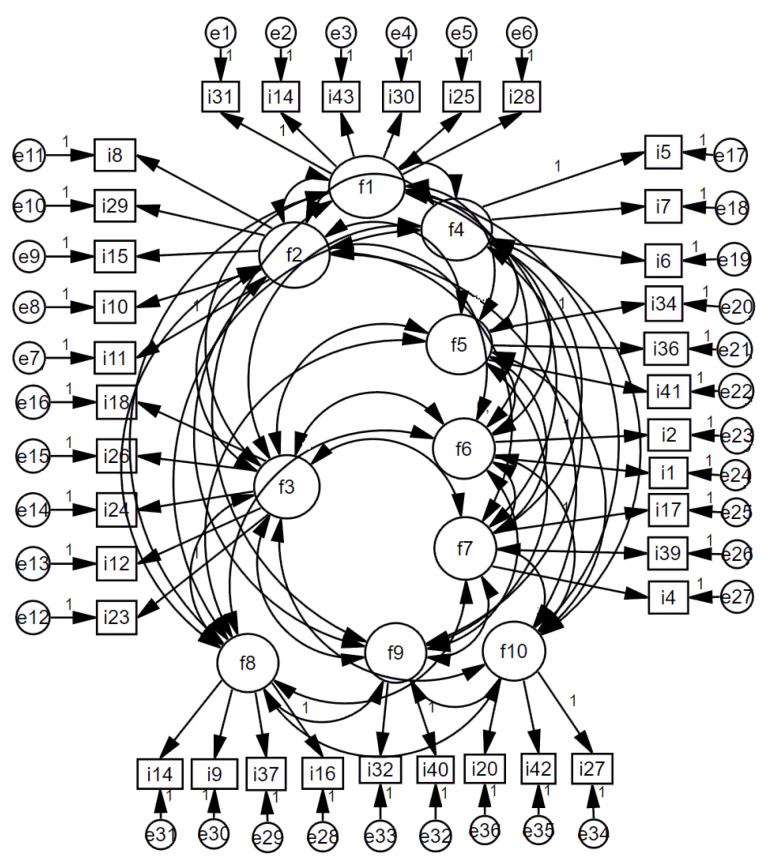
USHPS Scale Standardized Model.

**Figure 4 brainsci-13-00901-f004:**
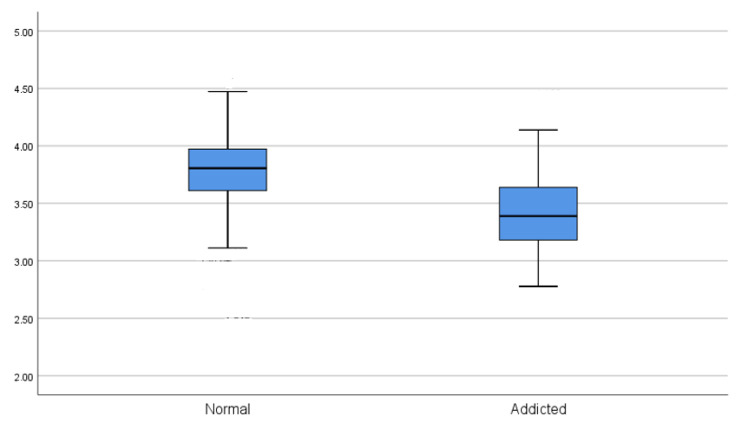
USHPS Scale Scores of the groups (the cut-off value was accepted as 2.5.).

**Table 1 brainsci-13-00901-t001:** USRAS Factor Structure and Explained Variance Ratio.

USRAS	Eigenvalue	Variance	Cummulative Varience
Factor 1	5.38	21.54	21.54
Factor 2	3.59	14.36	35.90
Factor 3	1.68	6.72	42.62
Factor 4	1.53	6.12	48.74
Factor 5	1.29	5.16	53.90
Factor 6	1.14	4.58	58.49

**Table 2 brainsci-13-00901-t002:** USRAS Item Factor Loads, Item Total Correlations, and Cronbach Alpha Values.

Factor	New Item Nu.	Items	Factor Load	Item Total Correlation	Cronbach Alpha
F1	1	Q6: I like the saying “Learn from the past, live in the present, look to the future”.	0.76	0.44	0.82
	2	Q11: It is good to think long term and I try to do that.	0.75	0.48	
	3	Q26: When I make a wrong decision, I go back and reevaluate.	0.73	0.40	
	4	Q3: It is always necessary to wait and be patient in order to reach a goal.	0.73	0.38	
	5	Q12: It is very important to be disciplined and to provide standards in life.	0.70	0.41	
	6	Q28: The most important thing that gives meaning to my life is the true meaning of death and the belief in eternity.	0.62	0.33	
	7	Q5: When I make a decision, I think 5–10 years ahead.	0.56	0.30	
F2	8	Q18: It is good to do hard work, but it is not for me.	0.80	0.38	0.80
	9	Q19: I am the type who cannot be bothered.	0.69	0.43	
	10	Q15: Getting pleasure is the important thing for me, hard work is not for me.	0.65	0.42	
	11	Q23: It is good to be principled and systematic, but it is not for me.	0.63	0.30	
F3	12	Q24: If there is an adventure, I forget almost everything.	0.86	0.63	0.82
	13	Q25: If there is a surprise, I will do it without thinking about the end.	0.82	0.53	
	14	Q23: If there is something different and attractive, I do it without thinking.	0.80	0.51	
	15	Q27: The purpose of life is to live in my own way.	0.50	0.34	
F4	16	Q2: People usually say I am in a hurry.	0.86	0.47	0.70
	17	Q31: I am known to be hasty and impatient, but I do not like this situation.	0.75	0.36	
	18	Q1: Getting what I want right away is very important to me.	0.71	0.40	
F5	19	Q4: If the problem is not resolved at home or in any case; alcohol, substance, or anything that gives pleasure suits me.	0.74	0.30	0.73
	20	Q34: When I cannot solve problems, I immediately turn to something that gives pleasure.	0.66	0.51	
	21	Q35: Hedonism does not end well, but I cannot give up on pleasurable things.	0.58	0.43	
	22	Q16: I find it difficult to express my feelings and turn to something that gives me pleasure.	0.52	0.39	
F6	23	Q29: My happiness is important instead of someone else’s happiness.	0.76	0.31	0.67
	24	Q8: My well-being and my own future come before the well-being of my family.	0.68	0.30	
	25	Q30: Those who say “The meaning of life; you have to be selfish to be happy after all.” are right.	0.65	0.30	
Total					0.72

As a result of the EFA, the USRAS scale form, consisting of 25 items and 6 factors, was rated in a 5-point Likert type as “Strongly disagree”, “Disagree”, “Neither/Nor Agree”, “Agree” and “Strongly agree”. A minimum of “1” and a maximum of “5” points can be taken from each item, and there are items that need to be scored in reverse. (According to “New Item Number”: 8, 9, 10, 12, 13, 14, 15, 16, 18, 19, 20, 21, 22, 23, 24, 25).

**Table 3 brainsci-13-00901-t003:** USRAS Average Scale Scores of the Groups.

Groups	X	SS	d
Non–SUD Female (n = 882)	3.83	0.42	
Non–SUD Male (n = 193)	3.76	0.45	
Non–SUD Total (n = 1075)	3.81	0.42	1.45 ^ab^
SUD Female (n = 6)	3.07	0.41	
SUD Male (n = 37)	3.22	0.43	
SUD Total (n = 43)	3.20	0.42	

The range of points that can be obtained is between 1 and 5. ^a^ The Non–SUD reference group is calculated as total X_1_ − X_2_/SD_Non–SUD_ ^b^ and the SUD reference group is calculated as total X_1_ − X_2_/SD_SUD._

**Table 4 brainsci-13-00901-t004:** USRAS Scale Scores by Harmful Substance Use.

Groups	X	SS	d
Non–SUD ^a^ “Never Use” (n = 767)	3.85	0.41	0.82 ^ac^; 0.24 ^ab^; 0.59 ^bc^; 0.57 ^cd^
Non–SUD ^b^ “Only Tobacco Use” (n = 214)	3.75	0.42	
Non–SUD ^c^ “Tobacco + Alcohol + Substance Use” (n = 53)	3.47	0.51	
Non–SUD Total (n = 1034)	3.81	0.42	
SUD ^d^ “Tobacco + Alcohol + Substance Use” (n = 43)	3.20	0.42	
SUD Total (n = 43)	3.20	0.42	

The range of points that can be obtained is between 1 and 5. ^a^ Non–SUD reference group is calculated as “Never Use” X_1_ − X_2_/SD_Non-SUD_
^b^ Non–SUD reference group is calculated as “Only Tobacco Use” X_1_ − X_2_/SD_Non–SUD_ ^c^ Non–SUD reference group is calculated as “Tobacco+ Alcohol + Substance Use” X_1_ − X_2_/SD_Non–SUD_ ^d^ SUD reference group is calculated as “Tobacco+ Alcohol + Substance Use” X_1_ − X_2_/SD_SUD._

**Table 5 brainsci-13-00901-t005:** USHPS Factor Structure and Explained Variance Ratio.

USHPS	Eigenvalue	Variance	Cumulative Variance
Factor 1	6.37	17.70	17.70
Factor 2	2.78	7.72	25.43
Factor 3	2.14	5.95	34.38
Factor 4	1.89	5.25	36.64
Factor 5	1.44	4.01	40.65
Factor 6	1.27	3.54	44.19
Factor 7	1.17	3.26	47.46
Factor 8	1.11	3.09	50.55
Factor 9	1.06	2.94	53.50
Factor 10	1.02	2.85	56.36

**Table 6 brainsci-13-00901-t006:** USHPS Item Factor Loads, Item Total Correlations, and Cronbach Alpha Values.

Factor	New Item Nu.	Items	Factor Load	Item Total Correlation	Cronbach Alpha
F1	1	Q31: My habits take up most of my time and my thoughts, but I am determined to complete them.	0.68	0.34	0.74
	2	Q44: I want to be careful, planned and act with the end in mind, but I cannot.	0.68	0.44	
	3	Q43: I do something that is harmful to me, and then I regret it.	0.68	0.32	
	4	Q30: My bad habits affect my health from time to time, I try to curb them.	0.67	0.40	
	5	Q25: I cannot plan; I got into a lot of trouble because of being hasty and impatient.	0.55	0.41	
	6	Q28: People say that I make a quick decision and act without thinking about the end, but I try hard to fix this.	0.55	0.35	
F2	7	Q8: Alcohol or drugs can be used for curiosity.	0.66	0.33	0.68
	8	Q29: I think pleasurable substances such as alcohol and drugs are harmless when taken in small amounts.	0.63	0.43	
	9	Q15: I think life is empty if there is no excitement, pleasure, and enjoyable alcohol substance.	0.58	0.33	
	10	Q10: When I feel troubled and distressed, the first thing that comes to my mind is to drink or seek something pleasant.	0.58	0.57	
	11	Q11: I do not think about the end when I am in a bad mood, I turn to whatever gives me pleasure.	0.55	0.73	
F3	12	Q18: It is very wrong to say that “Using tobacco and alcohol is a symbol of masculinity”.	0.65	0.35	0.82
	13	Q26: It is necessary not to drive in traffic, if it is very dangerous. I mostly achieve this.	0.63	0.37	
	14	Q24: I find social activities related to bad habits correct.	0.61	0.62	
	15	Q12: I think it is harmful to relax by drinking or taking something at that moment.	0.55	0.71	
	16	Q23: I do not want my children to use alcohol even if I use it.	0.54	0.63	
F4	17	Q5: Being curious often got me in trouble.	0.74	0.33	0.77
	18	Q7: I cannot control my curiosity, it has become a habit, it is harmful.	0.70	0.65	
	19	Q6: Curiosity is what drives me the most.	0.70	0.63	
F5	20	Q34: I get very bad if I do not drink alcohol or use substance.	0.83	0.84	0.70
	21	Q36: I cannot sleep if I do not drink alcohol or use substance.	0.81	0.44	
	22	Q41: Even though I know that it hurts me, I cannot get away from exciting and pleasurable substances.	0.53	0.36	
F6	23	Q2: For me, the more excitement, the more success. I do not think it does any harm.	0.83	0.61	0.82
	24	Q1: I think, “Life has no meaning, if there is no excitement”.	0.82	0.53	
F7	25	Q17: The rule of “First my interest, then my close circle.” is always valid.	0.78	0.60	0.76
	26	Q39: I think a person who does not think of his/her own interest first is an idiot.	0.76	0.42	
	27	Q4: I say “First me, then others”. I live by my mind.	0.60	0.39	
F8	28	Q14: When I am not happy, I do not feel incomplete, I can somehow relax myself.	0.74	0.40	0.86
	29	Q9: There are not many times in my life when I feel troubled and sad.	0.69	0.80	
	30	Q37: No matter what anyone says, I am a special and important person, I like myself.	0.50	0.70	
	31	Q16: It is very pleasant to do things that satisfy me, but I think you have to put up with the trouble.	0.50	0.32	
F9	32	Q32: I adjust the dose of alcohol or substance, I use it in a controlled manner.	0.77	0.62	0.80
	33	Q40: I am not addicted, I quit whenever I want.	0.76	0.40	
F10	34	Q20: I do not think it is harmful to eat a lot, drive dangerously or live fast.	0.65	0.33	0.78
	35	Q42: No need to think about harmful or dangerous situations, nothing will happen to me.	0.54	0.65	
	36	Q27: I cannot read or work for a long time.	0.50	0.32	
Total					0.83

As a result of the EFA, the USHPS scale form, consisting of 36 items and 10 factors, was rated in a 5-point Likert type as “Strongly disagree”, “Disagree”, “Neither/Nor Agree”, “Agree” and “Strongly agree”. A minimum of “1” and a maximum of “5” points can be taken from each item, and there are items that need to be scored in reverse. (According to “New Item Number”: 5, 7, 8, 9, 10, 11, 17, 18, 20, 21, 22, 23, 24, 25, 26, 27, 30, 32, 33, 34, 35, 36).

**Table 7 brainsci-13-00901-t007:** USHPS Average Scale Scores of the Groups.

Groups	X	SS	d
Non–SUD Female (n = 882)	3.78	0.27	
Non–SUD Male (n = 193)	3.75	0.30	
Non–SUD Total (n = 1075)	3.78	0.28	1.43 ^ab^
SUD Female (n = 6)	3.34	0.38	
SUD Male (n = 37)	3.45	0.33	
SUD Total (n = 43)	3.43	0.33	

The range of points that can be obtained is between 1 and 5. ^a^ The Non–SUD reference group is calculated as total X_1_ − X_2_/SD_Non–SUD_ ^b^ and the SUD reference group is calculated as total X_1_ − X_2_/SD_SUD._

**Table 8 brainsci-13-00901-t008:** USHPS Scale Scores by Harmful Substance Use.

Groups	X	SS	d
Non–SUD ^a^ “Never Use” (n = 767)	3.81	0.25	1.02 ^ac^; 0.28 ^ab^; 0.69 ^bc^; 0.24 ^cd^
Non–SUD ^b^ “Only Tobacco Use” (n = 214)	3.73	0.30	
Non–SUD ^c^ “Tobacco + Alcohol + Substance Use” (n = 53)	3.51	0.33	
Non–SUD Total (n = 1034)	3.78	0.28	
SUD ^d^ “Tobacco+ Alcohol + Substance Use” (n = 43)	3.43	0.33	
SUD Total (n = 43)	3.43	0.33	

The range of points that can be obtained is between 1 and 5. ^a^ Non–SUD reference group is calculated as “Never Use” X_1_ − X_2_/SD_Non-SUD_
^b^ Non–SUD reference group is calculated as “Only Tobacco Use” X_1_ − X_2_/SD_Non–SUD_ ^c^ Non–SUD reference group is calculated as “Tobacco + Alcohol + Substance Use” X_1_ − X_2_/SD_Non–SUD_ ^d^ SUD reference group is calculated as “Tobacco + Alcohol + Substance Use” X_1_ − X_2_/SD_SUD._

**Table 9 brainsci-13-00901-t009:** Pearson Correlation Values of Scales.

Scales	N	X	r	*p*
USRAS & USHPS	1134	3.79	0.54	0.000
	1134	3.76		

## Data Availability

Please contact the email nevzat.tarhan@uskudar.edu.tr to get the access of data.

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
