# Peer review of "Measuring the Awareness Levels of Individuals with Alcohol and Substance Use Disorders: Tertiary Prevention Standards and Development of Uskudar Result Awareness and Harm Perception Scales"

_brainsci, 2023, doi:10.3390/brainsci13060901_

Round 1

Reviewer 1 Report (Previous Reviewer 2)

The re-reviewed manuscript meets the criteria for a scientific article. With corrections and additions, the text reads completely differently. I suppose that the opinions of other reviewers also helped. I am pleased to say that in this form it is a good and valuable text.

Author Response

Thank you for your reviews. It was very useful for us and we had a chance to rework and edit the article. 

Reviewer 2 Report (Previous Reviewer 1)

Authors still show a conceptually flawed view on addictive diseases. It seems their statements come from different sources and are put together around a generic idea of "all in", where hierarchical orders and the role of different interventions is often misunderstood. For instance: "and thus the medical perspective that blames the person and claims that 160 the source of the problems is himself/herself alone is rejected" does not mirror what medical treatment of addiction is, at all. Also, the statement that psychosocial treatment within a medical maintenance program is the basis for relapse prevention is not true. 

The definition of substance abuse is flawed "Substance 59 use disorder can be defined as the desire to take a substance continuously or to reach the 60 substance at an intense level, in order to feel the pleasurable effects of a substance with a 61 drug quality and to avoid the discomfort caused by its absence."

Some statements are, indeed, not understandable at all, such as (about substance use disorder) "It is considered as a brain disease that has psychological components such as epilepsy, Parkinson’s and Alzheimer’s diseases and it significantly inhibits the functionality of the individual [5-8]."

The aim of the study is not fully clear: "In this study, it was aimed to develop two  scales suitable for a positive psychology-focused biopsychosocial perspective and to give  ideas about rehabilitation standards by determining awareness in individuals with or  without alcohol and substance use disorder." Authors state they are developing scales suitable for a theoretical approach, instead of focusing on the capacity to measure some parameters. This would "give ideas" about rehabilitation standards, but on what basis ? Moreover, they also speak about personalized treatment as an eventual benefit of awareness assessment: so personalization or standards ? In the closing remarks, they point out that the main goal of the study is to build up a scale and validate it, which does not implicate any advantage for treatment itself.

All in, the basic idea of awareness as a potentially influencing variable in the treatment of alcoholics is a valid one, and a scale is welcome.

Unfortunately, the paper must be re-written, focusing on the scale itself, and comparing it to existing instruments maybe. Before hypothesizing its role in treatment, concepts about addiction and its treatment should be amended.

see above

Author Response

Response to Reviewer Comments

Comment: Authors still show a conceptually flawed view on addictive diseases. It seems their statements come from different sources and are put together around a generic idea of "all in", where hierarchical orders and the role of different interventions is often misunderstood. For instance: "and thus the medical perspective that blames the person and claims that 160 the source of the problems is himself/herself alone is rejected" does not mirror what medical treatment of addiction is, at all. Also, the statement that psychosocial treatment within a medical maintenance program is the basis for relapse prevention is not true. 

Response: The biopsychosocial model’s importance is kept but the criticism towards the medical treatment of addiction is deleted from the article.

Comment: The definition of substance abuse is flawed "Substance 59 use disorder can be defined as the desire to take a substance continuously or to reach the 60 substance at an intense level, in order to feel the pleasurable effects of a substance with a 61 drug quality and to avoid the discomfort caused by its absence."

Response: This part is deleted from the article.

Comment: Some statements are, indeed, not understandable at all, such as (about substance use disorder) "It is considered as a brain disease that has psychological components such as epilepsy, Parkinson’s and Alzheimer’s diseases and it significantly inhibits the functionality of the individual [5-8]."

Response: The text “It is defined as an inappropriate use of behavior pattern characterized by repetitive social or interpersonal problems and the emergence of negative experiences at the clinical level. Substance use disorder can be defined as the desire to take a substance continuously or to reach the substance at an intense level, in order to feel the pleasurable effects of a substance with a drug quality and to avoid the discomfort caused by its absence. It is considered as a brain disease that has psychological components such as epilepsy, Parkinson’s and Alzheimer’s diseases and it significantly inhibits the functionality of the individual [5-8].” instead, this following sentence is added to the article:

Like epilepsy, Parkinson's, and Alzheimer's, substance use disorder is a brain-related disease that negatively affects a person's functionality in daily life [5-8].

Comment: The aim of the study is not fully clear: "In this study, it was aimed to develop two  scales suitable for a positive psychology-focused biopsychosocial perspective and to give  ideas about rehabilitation standards by determining awareness in individuals with or  without alcohol and substance use disorder." Authors state they are developing scales suitable for a theoretical approach, instead of focusing on the capacity to measure some parameters. This would "give ideas" about rehabilitation standards, but on what basis ? Moreover, they also speak about personalized treatment as an eventual benefit of awareness assessment: so personalization or standards ? In the closing remarks, they point out that the main goal of the study is to build up a scale and validate it, which does not implicate any advantage for treatment itself.

Response: The text  “The aims of the study include the development of result awareness and harm perception scales. And thus it was determined and compared the awareness levels in non-SUD group and substance use disorder (SUD) group. Measuring awareness levels of individuals with alcohol and substance use disorder will give information for following up after the treatment for the experts on the field.” has been changed into this:

In this study, it was aimed to develop two scales that can give ideas about rehabilitation standards by determining awareness in individuals with or without alcohol and substance use disorder. By so, experts on field can have information about the risk status of their patients in the follow-up process of rehabilitation, with the data obtained from the harm perception and result awareness dimensions in the scales.

Round 2

Reviewer 2 Report (Previous Reviewer 1)

ok, amending is not fully satisfactory but is fair.

This manuscript is a resubmission of an earlier submission. The following is a list of the peer review reports and author responses from that submission.

Round 1

Reviewer 1 Report

The theoretical premises of the paper are somewhat faulty. Addicted people are by definition ambivalent on a behavioral ground, and have poor insight, which means they have a low lever of awareness, if any at all. Thereby, results are far expected. 

Rehabilitation is definitely the ultimate goal of treatment, but it is not treatment itself. Relapsing into substance use in addicted individuals depends on addiction. Awareness of harm is irrelevant, since addicted people are aware of their disfunctional use, although they are unaware of its nature, mechanism and core issues (chronicity, automatic relapsing, autonomy with respect to environment). The statement that improving awareness is a way to prevent relapse is true, as long as this relies on ongoing treatment (when available). Beyond that, it is a failing viewpoint, mistaking the symptom for the method of treatment. Those who are healed, develop awareness of their disease, possibly (not of the toxic effects of the substance, or the wrong ways or reasons to resort to them).

Therefore, concluding that measuring awareness grants with some new instrument for prevention is itself unjustified, and conceptually faulty. Moreover, awareness of detoxified addicts is usually misleading, which means they are aware of wrong notions, they stick to wrong convictions, leading them to stop treatment, wean off methadone, rely on strengh of will, use rehabilitative contexts to prevent relapse.

The paper is a fair study on awareness in addicted people. The study design and its aim should be revised.

Author Response

Response to Reviewer 1 comments

The theoretical premises of the paper are somewhat faulty. Addicted people are by definition ambivalent on a behavioral ground, and have poor insight, which means they have a low level of awareness, if any at all. Thereby, results are far expected. 

Response: The development of two new scales was aimed in this study and the obtained results showed that scales are compatible with the literature. They support the literature, therefore being useful. It is natural to get expected results. Instead of the result, the focus should be on the scale materials that can be included in the post-treatment process.

Rehabilitation is definitely the ultimate goal of treatment, but it is not treatment itself. Relapsing into substance use in addicted individuals depends on addiction. Awareness of harm is irrelevant, since addicted people are aware of their disfunctional use, although they are unaware of its nature, mechanism and core issues (chronicity, automatic relapsing, autonomy with respect to environment). The statement that improving awareness is a way to prevent relapse is true, as long as this relies on ongoing treatment (when available). Beyond that, it is a failing viewpoint, mistaking the symptom for the method of treatment. Those who are healed, develop awareness of their disease, possibly (not of the toxic effects of the substance, or the wrong ways or reasons to resort to them).

Response: We agree with you in this view. We have made the necessary revisions and we agree with you. We specified the ongoing treatment process and that measuring self-awareness can be helpful after the treatment is finished. “When the treatment is finished, measuring the awareness levels can give the experts on field an idea based on data that is taken from the scales of this study.”

Therefore, concluding that measuring awareness grants with some new instrument for prevention is itself unjustified, and conceptually faulty. Moreover, awareness of detoxified addicts is usually misleading, which means they are aware of wrong notions, they stick to wrong convictions, leading them to stop treatment, wean off methadone, rely on strengh of will, use rehabilitative contexts to prevent relapse.

Response: We agree with you in this view. We have made the necessary revisions. We highlighted that measuring awareness levels can help expert on field to follow up the patients after the treatment process is over. By so, the scales are useful in the post-treatment process.

The paper is a fair study on awareness in addicted people. The study design and its aim should be revised.

Response: The aim of the study is revised: “The aim of the study includes the development of result awareness and harm perception scales. Thus, it was determined and compared the awareness levels in non-SUD group and substance use disorder (SUD) group. Measuring awareness levels of individuals with alcohol and substance use disorder will give information for following up after the treatment for the experts on the field.”

Reviewer 2 Report

Thank you for the opportunity to review this manuscript. After reading it carefully, there is a lot of ambivalence in me toward the content I have read. 

Let me start with a criticism.

I don't know why the authors refer to the DSM-4, when the DSM-5 classification has been used since 2013. Later on I see that they refer to the current version, but why in the introduction to the outdated one?

When describing the axial symptoms of addiction, I suggest using the DSM-5 or ICD-11 classification, which are the most correct sources of diagnostic criteria for the addiction syndrome. 

The texts from lines 117-127 would fit better in the discussion section. 

Likewise, the content from lines 128-138 doesn't really fit here either. As a reader, I wonder if they relate to the research presented? If so, it should be moved to the Materials and Methods section.

Lines 174 to 204 would also look better in the Discussion section, but the Discussion section was not included at all. That's probably why there is so much content strewn about in different places. In my opinion, this should definitely be changed.

Similarly, further pages up to the Material and Methods section. 

The introduction to the article is very long-winded and at this stage may bore the potential reader. As an addiction specialist, I was unable to understand what the authors were referring to.

Very much in the text phrases considered stigmatizing. The terms "addicted" have not been used for many years now, but rather "person suffering from addiction." Besides, the division into "normal" and "addicted" group is also hurtful. It sounds a bit like a group of "normal" and "abnormal" - called addicted.

The selection of respondents for the study is also unclear to me. While I have no objections to the group of people suffering from addiction, the other respondents - from whom data was collected via the Internet - are, in my opinion, too much of a random representation. I propose to be a little more specific about the sampling criteria.

The manuscript "defends" itself with good methods used in the research and interesting results, so I think the reviewed content has potential. However, I suggest a major overhaul of some of the sections, especially the addition of a Discussion section, where many excerpts from the introduction can be included.

I would be happy to review the manuscript after revisions.

Author Response

Response to Reviewer 2 Comments

Thank you for the opportunity to review this manuscript. After reading it carefully, there is a lot of ambivalence in me toward the content I have read. 

Let me start with a criticism.

I don't know why the authors refer to the DSM-4, when the DSM-5 classification has been used since 2013. Later on I see that they refer to the current version, but why in the introduction to the outdated one?

Response: We wanted to let the reader know the differences between DSM-4 and DSM-5 about addiction types. This difference is highlighted in the article.

When describing the axial symptoms of addiction, I suggest using the DSM-5 or ICD-11 classification, which are the most correct sources of diagnostic criteria for the addiction syndrome. 

Response: After this, the specified axial symptoms are included in the article based on DSM-5 rather than DSM-4.

The texts from lines 117-127 would fit better in the discussion section. 

Response: In order to present the literature part as a whole, these parts had to be dealt with in detail.

Likewise, the content from lines 128-138 doesn't really fit here either. As a reader, I wonder if they relate to the research presented? If so, it should be moved to the Materials and Methods section.

Response: In order to present the literature part as a whole, these parts had to be dealt with in detail.

Lines 174 to 204 would also look better in the Discussion section, but the Discussion section was not included at all. That's probably why there is so much content strewn about in different places. In my opinion, this should definitely be changed.

Response: Thank you for this feedback. As new scales were developed and offered a new perspective, the subject had to be dealt with in detail and the literature had to be presented comprehensively.

Similarly, further pages up to the Material and Methods section. 

Response: In order to present the methods section as a whole, these parts had to be dealt with in detail.

The introduction to the article is very long-winded and at this stage may bore the potential reader. As an addiction specialist, I was unable to understand what the authors were referring to.

Response: Within the scope of scale development, it was necessary to deal with the issue in detail and to present the literature comprehensively. Past scales had to be sampled. As the literature information presented is quite essential, we cannot exclude it from the article for now.

Very much in the text phrases considered stigmatizing. The terms "addicted" have not been used for many years now, but rather "person suffering from addiction." Besides, the division into "normal" and "addicted" group is also hurtful. It sounds a bit like a group of "normal" and "abnormal" - called addicted.

Response: The name of division between the normal/addicted group was changed in the study. Normal group was changed into non-substance use disorder (non-SUD) group. Addicted group was changed into substance use disorder (SUD) group. Substance Addicted People was changed into Individuals with Substance Use Disorder. This feedback was very important for us. All of the stigmatizing phrases was changed after this in the study.

The selection of respondents for the study is also unclear to me. While I have no objections to the group of people suffering from addiction, the other respondents - from whom data was collected via the Internet - are, in my opinion, too much of a random representation. I propose to be a little more specific about the sampling criteria.

Response: The sample was created by simple random sampling technique. We separated and compared those who got high scores from the scale and those who did not. Thus, randomized group was necessary and suitable for the purpose of the study. For the non-SUD group, additional data were collected from patients hospitalized in the AMATEM clinic. We tried to provide random data from both groups. We plan to focus on more specific groups in our next work.

The manuscript "defends" itself with good methods used in the research and interesting results, so I think the reviewed content has potential. However, I suggest a major overhaul of some of the sections, especially the addition of a Discussion section, where many excerpts from the introduction can be included.

Response: The research offers a new perspective as we have developed two new scales. With these, we can track self-awareness during post-treatment follow-up. We will pay attention to your suggestion when we conduct new research with scales in future studies. We especially valued your feedback on the crucial “stigmatizing” parts and changed them all. Thank you for your feedback.

I would be happy to review the manuscript after revisions.